# Beyond Analgesia: Psychobiotics as an Adjunctive Approach to Pain Management in Gastrointestinal Oncology—A *Post Hoc* Analysis from the *ProDeCa* Study

**DOI:** 10.3390/nu17172751

**Published:** 2025-08-25

**Authors:** Georgios Tzikos, Alexandra-Eleftheria Menni, Helen Theodorou, Eleni Chamalidou, Ioannis M. Theodorou, George Stavrou, Anne D. Shrewsbury, Aikaterini Amaniti, Anastasia Konsta, Joulia K. Tsetis, Vasileios Grosomanidis, Katerina Kotzampassi

**Affiliations:** 1Department of Surgery, Aristotle University of Thessaloniki, 54636 Thessaloniki, Greece; tzikos_giorgos@outlook.com (G.T.); alexmenn@auth.gr (A.-E.M.); a_shrewsbury@yahoo.com (A.D.S.); 2Department of Sociology, School of Social Sciences, University of Crete, 74100 Rethymno, Greece; psy5057@psy.soc.uoc.gr; 3Outpatient Surgical Oncology Unit, Chemotherapy Department, AHEPA University Hospital, Aristotle University of Thessaloniki, 54636 Thessaloniki, Greece; eleniham@gmail.com; 4Hellenic Institute for the Study of Sepsis (HISS), 11528 Athens, Greece; itheodorou@med.uoa.gr; 5Department of Surgery, 417 NIMTS (Army Share Fund Hospital), 11521 Athens, Greece; stavgd@gmail.com; 6Department of Anesthesiology and Intensive Care, School of Medicine, Aristotle University of Thessaloniki, 54124 Thessaloniki, Greece; aamaniti@auth.gr (A.A.); vgrosoma@auth.gr (V.G.); 7First Department of Psychiatry, “Papageorgiou” General Hospital of Thessaloniki, Aristotle University of Thessaloniki, 54124 Thessaloniki, Greece; konstaa@auth.gr; 8Uni-Pharma S.A., 14564 Athens, Greece; jtsetis@uni-pharma.gr

**Keywords:** psychobiotics, gastrointestinal cancer, cancer pain, chemotherapy, gut–brain axis, *post hoc* analysis, McGill Pain Questionnaire, adjunctive therapy, non-opioid analgesia

## Abstract

**Background**: Pain is a multifaceted and debilitating symptom in patients with gastrointestinal cancer, especially those undergoing surgical resection followed by chemotherapy. The interplay between inflammatory, neuropathic, and psychosocial components often renders conventional analgesia insufficient. Psychobiotics—probiotic strains with neuroactive properties—have recently emerged as potential modulators of pain perception through neuroimmune and gut–brain axis pathways. **Methods**: This *post hoc* analysis is based on the *ProDeCa* randomized, placebo-controlled trial, which originally aimed to assess the psychotropic effects of a four-strain psychobiotic formulation in postoperative gastrointestinal cancer patients receiving chemotherapy. In the current analysis, we evaluated changes in pain perception among non-depressed and depressed participants, who received either psychobiotics or placebo, along with standard analgesic regimes. Pain was assessed at baseline, after a month of treatment, and at follow-up, 2 months thereafter, using the Short-Form McGill Pain Questionnaire (SF-MPQ), capturing both sensory and affective components, as well as with the Present Pain Intensity and the VAS scores. **Results**: Psychobiotic-treated participants—particularly the non-depressed ones—exhibited a significant reduction in both quantitative and qualitative pain indices over time compared with placebo-treated ones. Improvements were noted in total pain rating index scores, sensory and affective subscales, and present pain intensity. These effects were sustained up to 2 months after intervention. In contrast, placebo groups demonstrated worsening in pain scores, probably influenced by ongoing chemotherapy and disease progression. The analgesic effect was less pronounced but still observable in the subgroup with symptoms of depression. **Conclusions**: Adjunctive psychobiotic therapy appears to beneficially modulate pain perception in gastrointestinal oncology patients receiving chemotherapy, with the most pronounced effects being in non-depressed individuals. These findings suggest psychobiotics as a promising non-opioid add-on for comprehensive cancer pain management and support further investigation in larger pain-targeted trials.

## 1. Introduction

Pain is a common and debilitating symptom among patients with gastrointestinal [GI] cancer, particularly those undergoing surgical resection followed by chemotherapy. The etiology of pain, recently re-defined by the International Association for the Study of Pain as “an unpleasant sensory and emotional experience associated with, or resembling that associated with, actual or potential tissue damage” [1]—in this population—is multifactorial, involving tumor biology, surgical trauma, and the cytotoxic effects of adjuvant therapies. Despite advancements in oncologic care, pain remains under-recognized and under-treated, significantly impacting the quality of life, emotional well-being, and treatment adherence of patients [2,3,4].

Surgical resection, while potentially curative, often results in acute postoperative pain due to tissue injury, nerve damage, and inflammation [5,6,7]. In a significant proportion of patients, this pain shifts toward a chronic postoperative pain, defined as pain persisting beyond the expected healing period of 3 months. Studies indicate that up to 30–50% of patients experience persistent pain following major abdominal surgeries, including those for GI malignancies [8,9]. Moreover, neuropathic pain components may arise due to the involvement of somatic or visceral nerve plexuses affected during lymphadenectomy or aggressive dissection during extensive resections [7,10].

Chemotherapy further exacerbates the pain burden through both direct and indirect mechanisms, such as peripheral neurotoxicity, myelosuppression-induced bone pain, and mucositis. Platinum-based compounds, namely, oxaliplatin and taxanes (paclitaxel) are known to induce chemotherapy-related peripheral neuropathy, manifesting as burning, tingling, or diffuse pain, affecting 30–60% of patients and often persisting long after treatment ends [11,12]. Additionally, mucositis, myalgias, and arthralgias are common adverse side effects that aggravate the overall pain experience. The chemotherapy-induced systemic inflammatory milieu sensitizes both peripheral and central nociceptive pathways, leading to enhanced pain perception [13,14].

Cancer pain, affecting nearly 66% of cancer patients [15], is not merely a somatic experience but a complex phenomenon involving emotional and psychosocial dimensions. According to Mendoza-Contreras et al. [16], there are three main types: the *sensory–discriminative*, which involves the quality, location, duration, and intensity of pain; the *motivational–affective*, which includes subjective facets such as suffering, aversion, dislike, and experienced emotional changes; and the *cognitive–evaluative*, which comprises earlier experiences and response strategies. From this point of view, the close relationship of emotional response to any sensory dimension, and vice versa, suggests that anxiety and depression, being common comorbidities in cancer patients, can significantly amplify pain perception through top–down modulation in the central nervous system [17,18]. This bio-psycho-social model of pain underscores the need for comprehensive assessment and multimodal management approaches that go beyond pharmacological approaches [5,6].

Validated pain assessment tools, such as the McGill Pain Questionnaire, the Brief Pain Inventory (BPI), the Numerical Rating Scale (NRS), and the Visual Analog Scale (VAS), have been used from the past in oncology settings to quantify mainly the pain intensity and interference with daily functioning [19]. These tools are instrumental in monitoring the progression or remission of pain symptoms over time, especially in response to therapeutic interventions. Furthermore, in order to assess the sensory-discriminative and the motivational–affective dimensions of pain, the Short-Form McGill Pain Questionnaire was developed, and has been widely used for almost 40 years as a well-recognized assessment tool in both clinical and research settings [20].

Unmanaged pain in patients with GI cancers has been associated with poor nutritional intake, sleep disturbances, reduced mobility, reduced or eliminated ability to work, no desire to socially interact, and an overall decline in performance status [21,22,23]. It is also a major determinant of patient-reported outcomes and satisfaction with care. Therefore, effective pain control is not only an ethical imperative and a medical obligation under the right to health [24] but also a clinical necessity to support optimal recovery and therapeutic efficacy [3]. Given the complexity and multifactorial nature of cancer-related pain, there is a growing interest in integrative approaches that modulate pain perception, including those involving neuroimmune and gut–brain axis pathways [25,26]. While pharmacological analgesia remains the cornerstone of pain control, complementary approaches targeting both physiological and psychological aspects of pain are gaining ground [5].

Emerging research has increasingly highlighted the potential role of probiotics having psychotropic properties—referred to as psychobiotics—in modulating pain perception, particularly in postoperative and chemotherapy settings. The mechanisms by which these agents influence pain are multifaceted and include anti-inflammatory effects, immunomodulation, enhancement of intestinal barrier integrity, and modulation of both the gut–brain axis and the neuroendocrine signaling [25,26,27]. Specific strains, such as *Lactobacillus acidophilus* NCFM and *L. rhamnosus* GG, have been shown to increase the expression of *μ*-opioid and cannabinoid receptors in intestinal epithelial cells, as well as in animal models, directly influencing nociceptive pathways and reducing visceral hypersensitivity [28].

Preclinical studies in a rat model of colonic dilatation have demonstrated that administration of certain probiotic strains increases pain thresholds and reduces visceral motor responses—effects comparable to or synergistic with low-dose morphine [29,30,31]. Moreover, psychobiotics such as *Lactiplantibacillus plantarum* PS128 have been found to alleviate pain via suppression of the pro-inflammatory cytokines IL-6, IL-1β, and TNF-α and via the modulation of the hypothalamic–pituitary–adrenal axis, often being dysregulated due to cancer-related stress and pain [27]. The analgesic properties of psychobiotics are particularly important in the postoperative setting, where both inflammatory and neuropathic pain coexist, and in the chemotherapy phase, where intestinal dysbiosis and mucosal damage often exacerbate gastrointestinal and systemic pain syndromes. Notably, psychobiotics also contribute to analgesia by producing neuroactive compounds, such as gamma-aminobutyric acid (GABA), serotonin, and short-chain fatty acids, specifically butyrate, all of which are involved in the regulation of sensory neuron excitability [25].

Although clinical trials remain few in number, improvements in abdominal pain, pain interference scores, and the overall symptom burden in cancer patients receiving adjunctive probiotic therapy have been reported [32]. Collectively, evidence supports further investigation of psychobiotics as a promising non-opioid adjunct in the multidisciplinary management with GI cancer-related pain in patients undergoing surgical excision plus chemotherapy. Recently, we performed a randomized placebo-controlled trial (the *ProDeCa* study) to investigate the effects of a specific psychobiotic regime to ameliorate depression status, as well as anxiety and stress parameters, in participants in a chemotherapy course for a newly operated GI malignancy [33].

In the present *post hoc* analysis, we aimed to investigate the potential analgesic effects of psychobiotics or placebo, given as a 1-month add-on treatment to classically prescribed analgesics. We mainly focused on the non-depressed participants who—at the beginning of the postoperative chemotherapy course—were self-assessed by means of the McGill Pain Questionnaire and found to be experiencing pain. In parallel, we assessed the corresponding group of depressed individuals—already allocated to psychobiotics or placebo, both sub-groups receiving standardized analgesic treatment.

## 2. Materials and Methods

### 2.1. Study Design and Participants

This *post hoc* analysis is based on data derived from the recently published *ProDeCa* study [33]. Eligible participants were adults, aged ≥ 18 years, on postoperative chemotherapy for GI oncology, after tumor resection, having the ability to provide informed consent, and with no cognitive impairment. Exclusion criteria also included active psychiatric disorders or a history of suicide attempts and not being on prescribed antidepressants or recent antibiotic or probiotic use within 4 weeks prior to enrollment and severe immunosuppression.

### 2.2. Depression Assessment and Study Groups

Upon study enrollment, and after participants were randomly allocated to the psychobiotics or placebo groups, they were assessed for the existence of depression symptoms by means of the 17-item Hamilton Depression Rating Scale (HDRS-D-17) and the Beck’s Depression Inventory (BDI-II) score. Participants were thus classified as depressed or non-depressed based on an established cut-off point score; specifically, an HDRS-D-17 total score of ≤7 was considered indicative of non-depression, while scores above this threshold indicated clinically relevant depressive symptoms [34]. For this *post hoc* analysis, we predominantly used the participants classified as non-depressed in order to specifically evaluate the effects of 1-month psychobiotics treatment or placebo on pain perception; for comparisons, the depressed groups—psychobiotics- or placebo-treated—were also used. For the better understanding of the effects of psychobiotics on pain perception, in this analysis, we followed each participant from baseline (T0) to treatment termination at T1 and then to the end of the 3rd month, follow-up, at T2.

### 2.3. Pain Evaluation

Pain perception was assessed using the Short-Form McGill Pain Questionnaire (MPQ-SF), a validated and widely used self-report instrument that measures both sensory and affective components of pain through a series of descriptive adjectives, although it does not employ established clinical cut-off points [20,35]. It consists of (i) the Pain Rating Index, which has a multidimensional profile, by adding sensory (11 items) and affective (4 items) dimensions of pain quality and intensity, rated on a Likert scale (0 = no pain, 1 = mild, 2 = moderate, and 3 = severe), the possible range of scores being from 0 to 45; (ii) a visual analog VAS scale (16th item) ranging from 0 (no pain) to 100 (worst possible pain); and (iii) an indicator of present pain intensity (PPI), the 17th item. The PPI scale measures pain experienced right now based on a 6-point Likert scale with choices of 0 = no pain, 1 = mild, 2 = discomforting, 3 = distressing, 4 = horrible, and 5 = excruciating. Assessments were conducted at baseline (prior to treatment initiation—T0); 4 weeks later, upon treatment termination (T1), and at the end of the 3rd month—at follow-up (T2).

We initially assessed the participants quantitively: We recorded the number of subjects in pain at baseline within each group (non-depressed and depressed plus psychobiotics or placebo) and the differentiation in the number at T1 and T2. We then assessed them qualitatively: each subject was recorded as presenting pain intensity at T0, T1, and T2; and finally, in a more detailed manner, we recorded the pain rating index at T0, T1, and T2, analytically considering the sensory and affective components of their pain at each time-point.

### 2.4. Chemotherapy and Pain Management Scheme—Protocols

All GI cancer patients receiving chemotherapy after tumor surgical resection were allocated to either the FOLFOX scheme (5-fluorouracil (5-Fu), leucovorin (folinic acid), and oxaliplatin) mainly for colorectal cancer affecting the majority of our study population or the FOLFIRI scheme (5-Fu, leucovorin and irinotecan), per protocol.

Regarding pain management protocols, all participants were closely monitored as outpatients at the Chronic Pain Management Unit of our university hospital and treated according to the WHO three-step pain ladder [36], which, for decades, has remained the keystone to cancer pain management. According to this scheme, NSAIDs and paracetamol (acetaminophen) constitute the first step of the WHO ladder and have also been added as important components of the second and even third step, with moderate-to-severe pain. Weak opioids—codeine or tramadol—plus paracetamol or tramadol alone in higher doses of up to 300 mg/day constitute the second step, where codeine and tramadol may be switched for better patient response. Finally, strong opioids (morphine, fentanyl, methadone) in combination with NSAIDs or paracetamol are introduced at the third step [37]. In Greece, fentanyl is preferred to morphine and is used in the form of dermal patches plus nasal or sublingual sprays in the case of paroxysmal pain attack. However, between weak opioids and a strong one, fentanyl, the new therapeutics oxycodone or tapentadol may be taken orally. Almost all the participants of the present study relied on step 2, except for 9 pancreatic cancer patients additionally taking tapentadol.

### 2.5. Intervention Protocol

Individuals allocated to the psychobiotic group, twice daily, received a formulation of four psychobiotics at a daily dose of 1.76 × 10^11^ CFU; the regime contained the following bacterial strains at the specified doses: *Bifidobacterium animalis* subsp. *lactis* LMG P-21384 (BS01) (2.50 × 10^10^ CFU), *Bifidobacterium breve* DSM 16604 (BR03) (1.00 × 10^10^ CFU), *Bifidobacterium longum* DSM 16603 (BL04) (8.00 × 10^9^ CFU), and *Lacticaseibacillus rhamnosus* ATCC 53103 (GG) (4.50 × 10^10^ CFU). The total duration of the intervention was 4 weeks, starting at the beginning of the chemotherapy course. The placebo sachets were identical in packaging, appearance, consistency, and solubility in drinking water, as well as in taste and smell, to ensure blinding. Compliance was assessed by capsule counts at the end of treatment.

### 2.6. Statistics

The normality of the data distribution was assessed by performing the Kolmogorov–Smirnov test. Continuous variables were presented as means ± standard deviation (SD) when normality was assumed or as median and interquartile range (IQR) when normality was violated. Categorical variables were presented as absolute numbers and proportions. For a comparison of means between independent samples, the Student *t*-test was applied when the data followed a normal distribution, whereas the Mann–Whitney U test otherwise. For related samples, the Wilcoxon signed-rank test was used for two-group comparisons, and the Friedman test for three consecutive assessments within the same group. For categorical data, the chi-square test was used. When subgroup comparisons were performed, Bonferroni correction was applied to adjust for multiple testing. Statistical significance was set at a two-tailed α level of 0.05. All analyses were performed using IBM SPSS Statistics for Windows, version 25.0 (IBM Corp., Armonk, NY, USA), while graphs were drawn with both the Microsoft Excel (Microsoft Office Professional Plus 2019) and the GraphPad Prism, version 10.5.0 for Windows (GraphPad Software, Boston, MA, USA).

## 3. Results

For this *post hoc* analysis, we used the participants of the *ProDeCa* study [33]—classified as non-depressed (HDRS-D-17 score ≤ 7) or depressed at baseline and then allocated to the psychobiotics or placebo groups—in order to evaluate the effects of 1-month psychobiotic treatment or placebo on pain perception. We specifically focused on the effectiveness of psychobiotics against placebo on cancer pain perception in the group of non-depressed individuals because it is, theoretically, a more mentally “stable” population. For comparisons, we referred to the matching depressed groups.

### 3.1. Demographics

In this *post hoc* analysis, we primarily referred to 167 non-depressed participants, already allocated to psychobiotics (n = 84) or to placebo (n = 83) groups. Regarding the other group of 99 depressed participants, there were 48 in the psychobiotics and 51 in the placebo groups (*p* = 0.775). Their baseline demographic characteristics and all other related information are tabulated in Appendix A.

### 3.2. Evaluation of the Number of Participants Experiencing Pain in Each Group

There was no statistically significant difference identified between the numbers of non-depressed and depressed participants experiencing pain, all self-assessing by means of the VAS scale: there were 58 out of the 167 (34.73%) and 41 out of the 99 (41.41%) (chi square = 1.188, *p* = 0.276).

When we further analyzed these pain-suffering participants with respect to the psychobiotic treatment they had been previously allocated, 27 out of 58 (46.55%) and 18 out of 41 (43.90%) were found in non-depressed and depressed groups, respectively. In the same manner, there were 31 out of 58 (53.45%) and 23 out of 41 in the non-depressed and depressed groups taking the placebo. In other words, there was no difference, at baseline, in the number (or the ratio) of participants suffering pain among each of the four groups (chi square = 0.068, *p* = 0.794) (Table 1).

### 3.3. Assessment of Pain Intensity Based on VAS Score

When the pain intensity was quantitatively analyzed using the Visual Analogue Scale (VAS—max 100 points), a statistically significant decline was found after the 1-month psychobiotics treatment in the non-depressed group (T0 to T1, *p* = 0.002), further declining progressively, beyond the treatment termination (T1 to T2 *p* = 0.001 and T0–T3 *p* < 0.001). Placebo-treated participants (non-depressed) presented a progressively significant increase in pain, obviously due to disease aggravation (T0–T2, *p* = 0.008) (Table 2).

Correspondingly, the depressed group taking the psychobiotics also presented a decrease in pain intensity at T1, *p* = 0.029, compared with T0, and, further on, at T2, *p* = 0.046, compared with T1, thus achieving a significant reduction (T0 to T2, *p* = 0.002). Placebo-treated participants (depressed) presented a small but statistically significant increase in pain intensity over time (*p* = 0.023 between T1 and T2, *p* = 0.035 between T0 and T2).

Based on all of the above, it is prominent that non-depressed individuals experience pain of higher intensity in comparison with depressed ones, independently of receiving either psychobiotics or placebo. This difference is clearly shown in Figure 1.

### 3.4. Present Pain Intensity Index

By analyzing the results obtained after self-assessment of the present pain intensity index (graded as 0—no pain; 1—mild; 2—discomfort; 3—distressing; 4—horrible; and 5—excruciating pain), we found a significant decrease in the groups of non-depressed and depressed participants who had taken psychobiotics and a significant increase in the groups of non-depressed and depressed participants who had taken placebo (Table 3, Figure 2).

### 3.5. Pain Rating Index

Finally, by calculating the total pain rating index, that is, the sum of the severity assessment (rating from 0 to 3) of each of the 15 items of the McGill short-form questionnaire, characterizing the pain quality at each time-point (T0, T1, and T2), we found the following deviations, both within the four groups and between them (Table 4).

The initial observations on these results are that (i) the non-depressed participants presented with a higher pain rating index in relation to the depressed group, irrespective of the treatment (psychobiotics or placebo); (ii) the psychobiotics significantly decreased the pain rating index over time in non-depressed participants, while a progressively, but not significant, decreasing index (less pain) was present in the depressed, psychobiotics-treated participants; and (iii) the placebo—independently of the mood status (non-depressed—depressed) of participants—kept the pain rating index stable during the treatment period, while presenting a significant increase (more pain) thereafter.

Next, we analyzed the pain rating index of each of the 15 items—descriptors, which are presented in Appendix A, per study group. Since it is known that the first 11 descriptors (items 1–11) refer to somatic or sensory pain and the other four (items 12–15) to emotional of affective pain, we then assessed them separately for each of the study groups. Thus, we identified the most common types of sensory pain (11 items) described by non-depressed participants at T0 to be throbbing, shooting, sharp, cramping, and gnawing, while stabbing, shooting, cramping, and gnawing were described by depressed participants. Regarding the affective dimension of pain (4 items), the most common was the fearful type in non-depressed and the sickening, fearful, and punishing/cruel types in depressed individuals. In both mood categories, the lowest proportion had the tender-type pain. Based on all of the above, it is clear again that non-depressed individuals expressed statistically higher pain scores in each one of the parameters—descriptors in relation to the depressed. Additionally, they used slightly different expressions to describe both the sensory and the affective pain in relation to depressed individuals. These differences are clearly shown in Figure 3a,b.

Then, we statistically assessed the above findings separated into the two types of pain, the sensory and the affective pain. It is clear that there was a statistically significant reduction in the sensory pain rating index (*p* = 0.04), as well as in the affective pain rating index (*p* < 0.001) in non-depressed participants who had taken psychobiotics. The same trend (reduction—but not significant) was observed in the sensory (*p* = 0.128) and the affective pain rating index (*p* = 0.28) in depressed participants who had taken psychobiotics.

However, there was a statistically significant increase in the sensory pain rating index (*p* < 0.001) and a trend towards an increase only in the affective pain rating index (*p* = 0.123) in non-depressed participants taking the placebo. The same trend (significant increase) was observed in the sensory (*p* = 0.03) and the affective pain rating index (*p* = 0.025) in depressed participants taking the placebo. The descriptive statistics (min, max, mean (SD), and median (IQR)) of both the pain-origin groups throughout the study periods are presented in Table 5, while the fluctuations of values in the pain rating index of each of the 11 sensory and the 4 affective descriptors at the three time-points per group of participants are presented in Figure 4a,b.

Finally, after psychobiotics or placebo treatment (T1), as well as at follow-up (T2) at 3 months from baseline (T0), a differentiation of the values of each descriptor is prominent, the sum of which expresses the total score of the pain rating index. The differences per item (increase, decrease, or remain stable) are illustrated in a spider graph (Figure 5a–d).

## 4. Discussion

In this *post hoc* analysis of patients who received psychobiotics or placebo after being operated on for gastrointestinal cancer and being on a chemotherapy course, we found that non-depressed ones—as determined by the HDRS-D-17 score—experienced a greater reduction in pain intensity, during and after psychobiotic treatment in comparison with placebo-treated ones. The same reduction trend, but to a much lower degree, and with no statistically significant differences between study time-points, was also observed in depressed individuals.

Cancer, of whatever etiology, is a multifactorial disease, directly associated with intense pain and fatigue. Furthermore, pain is the foremost distress experienced during cancer treatment, particularly as a consequence of surgical trauma and a major side effect of the subsequent chemotherapy [38,39]. Cancer-related pain involves a complex interplay of physiological, psychological, emotional, and social factors; additionally, spiritual components, gender, and even a preoperative pain history, pain fear, and pain sensitivity seem to ultimately have a significant impact on an individual’s pain and thus, finally, on their quality of life [40,41,42,43,44,45].

Despite the advancements and novelty in pharmacological approaches, the improvements in dosage monitoring being adjustable to personal needs [46,47], and the updated pain guidelines [48], the optimization of pain management remains a challenge [39,42]. Even today, it is an open secret that more than 70% of individuals with cancer experience the distressing symptom of pain, nearly half of which is inadequately controlled [15,49], and even more, some patients do not receive adequate pain medication [2,50,51,52]. At the same time, addiction to analgesics and the adverse events of pharmacological interventions in general also continue to pose critical challenges to pain management [53,54].

The European “Pain in Cancer” survey in adults reported that 56% of cancer patients experience moderate to severe pain (≥5 out of 10 on a numeric rating scale (NRS)) at least several days a month, while 27% rate their pain as severe (≥7 of 10 on NRS). Furthermore, only 24% of cancer patients receive strong opioids, and up to 11% of those with moderate to severe pain receive no analgesia at all. This inadequate pain management has led to distress in 67% of patients and the desire for death in 32% [55]. In the same manner, according to a recent meta-analysis, the prevalence of pain faced by cancer patients analyzed during the period 2014 to 2021 was found to be 44.5%, while the ratio of moderate to severe pain in the same population was 30.6% [56]. In the present analysis, we found, at baseline, that 99 out of the 266 participants (37.2%) had to live with some type and/or intensity of pain, the ratio being quite similar in both non-depressed and depressed participants, this occurring despite the adequate doses of analgesics taken, since all were routinely monitored by the medical staff of the Pain Management Clinic, as per protocol.

Emotional distress has been revealed to be significantly associated with pain perception; the higher the level of psychological distress, including depression, anxiety, hostility, mood disturbances, and anger, the higher the concomitant pain intensity [57,58]. In our cohort, when the participants were split according to their mood status—specifically depression or non-depression—41 out of the 99 depressed individuals (41.4%) as opposed to 58 out of the 167 (34.7%) reported various pain intensities. At a first glance, the difference is not so interesting. However, when the pain intensity was quantitatively analyzed by means of the Visual Analogue Scale, the maximum being 100 points, it was surprisingly found that the non-depressed had a higher intensity of pain, at a level of almost 10 points higher, in relation to the depressed; in other words, pain perception, contrary to general thoughts, was “depressed” in depressed participants, despite the fact that none received antidepressants (Table 2). Furthermore, when we calculated the pain rating index using the short-form 15-item questionnaire of McGill, we identified the median sensory pain score, being the sum of the first 11 items/descriptors, as only 7 (0 to 33) for the depressed participants in contrast with the non-depressed, which was 12 (0 to 33). Similarly, regarding the affective dimension of pain (4 items), the median score was 1.5 (0 to 12) in depressed and 4.5 (0 to 12) in non-depressed individuals. Based on these findings, we can assume that the depressed participants experienced dark moods, sadness, and uncertainty derived from their understanding of the progression of their illness, resulting in their giving up all interests in life and all efforts to fight the disease. This could be why they self-assessed with lower pain scores. This option is also further supported by the pain scores found during the study progress, which will be discussed a little below.

Besides the splanchnic pain due to gastrointestinal cancer itself and the postoperative pain, it is well accepted that more than half of such a population develops chemotherapy-induced peripheral neuropathy, caused by multiple chemotherapeutic drugs that can induce damage to sensory, motor, autonomic, or cranial nerves; such a neuropathy is perceived and commonly described as numbness, loss of proprioception sense, tingling, burning pins and needles, hyperalgesia, or an acute pain syndrome [12,59,60,61]. The chemotherapy-induced peripheral neuropathy is a prominent dose-limiting toxicity of either oxaliplatin or irinotecan given to cancer patients, all contained within the FOLFOX or FOLFIRI treatment schemes applied to our patients. Specifically, oxaliplatin, besides causing a dose-dependent chronic sensory neuropathy, is also able to trigger an acute sensory neuropathy, classically characterized by cold-induced paresthesia and muscle cramping sensations [61,62]. Furthermore, many of the post-treatment pain syndromes, including those related to surgical trauma—incidences of which vary relating to surgery type or even to the impact of muscle retractors [5,6,38,63]—have a component of neuropathic pain; Pogatzki-Zahn et al. [64] documented them using functional MRI changes in brain activation during incisional pain in healthy volunteers, affecting the secondary somatosensory cortex, the frontal cortex, and the limbic system. Neuropathic pain is clinically manifested as allodynia, that is, pain from innocuous stimuli, or hyperalgesia—exaggerated response to a common painful stimulus, alternatively, as spontaneous or continuous pain, described as paresthesia or dysesthesia, or lightning-like unprovoked pain, along with psychiatric comorbidities, which in turn enhance pain perception [65,66,67,68,69,70].

In recent years, several trials have emphasized the positive effects of probiotics on various aspects of mental and emotional state, from mood disorders to pain perception [71]. Early studies by Rousseaux et al. [28] on human HT-29 epithelial cells revealed the ability of *L. acidophilus* NCFM and *L. salivarius* Ls-33 to induce a sustained increase in OPRM1 mRNA (-opioid) expression and only of *L. acidophilus NCFM* to induce, additionally, CNR2 mRNA (cannabinoid) expression. Then, in a colonic distension rat model, the oral administration of *L. acidophilus NCFM* was found to decrease the normal visceral perception, allowing a 20% increase in pain threshold, or resulting in an anti-nociceptive effect of the same magnitude as that achieved by 0.1 mg/kg morphine, given subcutaneously. The authors thus concluded that the close contact of *L. acidophilus NCFM* with the gut epithelial cells is able to induce, via the NF-B pathway, opioid and cannabinoid receptors to mediate the normal perception of visceral pain, as also occurring with morphine [28]. Furthermore, in a more recent study, the probiotic regime of *L. helveticus* R0052 and *B. longum R0175*, both having psychotropic effects, was reported to attenuate the HPA axis-induced stress response [72], as also occurs with B. longum SX-1326 [73], which has also been found to alleviate irinotecan side effects by means of reversing gut dysbiosis, increasing the abundance of *Dehalobacterium*, *Ruminococcus,* and *Mucispirillum* in a colorectal cancer model in mice. At the same time, Wang et al. [74], in a murine model of chronic gut constriction injury, showed that a 15-probiotic regime can improve neuropathic pain and muscle atrophy, the improvement being attributed to the ability of probiotics to manage pain of inflammatory origin. Furthermore, their findings of probiotic-induced gut microbiota repair—mainly by reversing the significantly decreased abundance of *Akkermansia* and increased *Ruminococcaceae*, considered to be the cause of neuropathic pain—led them to suggest that probiotics could be utilized for treatment [74].

From this point of view, psychobiotics (a plethora of specific probiotic bacteria), since their introduction by Ted Dyan and John Cryan in 2013 [75,76], are well established due to their unique characteristics of producing and delivering neuroactive substances such as gamma-aminobutyric acid and tryptophan, which assist in serotonin synthesis, act on the brain–gut axis, and generally manifest beneficial effects in psychiatric illness, by means of modifying the disturbed microbiome abundance, on a strictly specific strain level [77,78,79,80]. Their psychotropic properties have already been established both experimentally and in humans [33,81,82,83,84], while the direct role of 5-HT in pain, and more specifically that of visceral origin, is also well documented [85,86,87,88].

In a recent randomized double-blind controlled clinical study [33], we found the a four-psychobiotic regime containing *Bifidobacterium animalis subsp. lactis* LMG P-21384 (BS01) (2.50 × 10^10^ cfu/dose), *Bifidobacterium breve* DSM 16604 (BR03) (1.00 × 10^10^ cfu/dose), *Bifidobacterium longum* DSM 16603 (BL04) (8.00 × 10^9^ cfu/dose), and *Lacticaseibacillus rhamnosus* ATCC 53103 (GG) (4.50 × 10^10^ cfu/dose), administered for a month against placebo twice daily (total daily dose of 1.76 × 10^11^ cfu), resulted in a significant improvement in depression status, as well as in anxiety and stress levels. The present *post hoc* analysis of data from the previous study [33] revealed that, indeed, psychobiotic treatment significantly reduced the pain intensity assessed by the VAS score, as well as the present pain intensity index and the total pain rating index (McGill questionnaire, short form) after a month of treatment, only in the non-depressed participants; depressed participants, having at baseline a significantly lower pain perception than non-depressed ones, presented only a reduction trend with no statistical difference. For comparison, both groups taking placebo exhibited a progressive reduction in pain levels, clearly attributable to disease exacerbation.

The presence of a significantly lower pain intensity index and total pain rating index in the depressed participants versus the non-depressed ones—either receiving psychobiotics or placebo—sounds strange and questionable, since it is well recognized that a balanced mental state copes better with all stressful stimuli. The theory of “gate-control” might explain it, since its ultimate conclusion is that the pain process can be mediated by changing the way a person cognitively processes his/her pain experience [89]. Normally, the pain can be transmitted to the patient’s brain through nerves and activates the relevant prefrontal cortex and other regions, which could affect the normal emotion regulation when the region receives pain signals. The “gate”, located somewhere within the brain, may be opened or closed, its situation determining the amount of pain the individual experiences. The pain message originates at the point of aggravation, the signal is transmitted to the brain, and the pain is then perceived by the person only if the gate is open. The use of coping strategies may close the gate, meaning that the brain will not recognize or give credence to the pain signal; or the gate may be opened, bringing the pain signal to the brain’s awareness, meaning that the patient focuses on thoughts of pain [89].

Based on or affected by the “gate-control theory”, other authors further analyze and explain the findings either as due to perception and reporting bias or simply as due to differences in coping mechanisms. Depressed individuals may underreport their physical symptoms, including pain, due to general apathy, emotional blunting, or reduced interoceptive awareness that often accompanies depressive states. Others suggest that depressed individuals may develop pain tolerance or resignation as a coping mechanism, as opposed to non-depressed individuals who might be more reactive or attentive to pain sensations [90]. This option is also supported by the biochemical theory of depression, which suggests that a neurochemical imbalance of monoamines, as is serotonin, underlies depression, and since these neurotransmitters are likely to play a key role in modulatory pain pathways, this may lead to altered pain perception [91]. Additionally, a preliminary meta-analysis of 6 experimental studies [92] reported decreased pain in depressed groups relative to healthy control groups. This finding has become more significant after a new meta-analysis—the largest and most comprehensive to date—of 25 studies (567 depressed and 587 healthy participants) investigating responses to experimentally induced pain, which revealed a higher overall pain threshold, meaning reduced pain in the depressed in comparison with healthy participants [91].

The present *host hoc analysis* has some limitations: First of all, pain perception is a totally subjective experience, which means that each participant may subjectively sense, experience, and interpret the meaning of his/her pain uniquely; thus, we, as physicians, have no other choice but to rely on patient self-reports. Therefore, the assessment of pain at baseline and its response to the analgesic treatment protocol is totally affected by the patient’s emotional status, probably beyond the pain perception itself. Second, the present study is a *post hoc* analysis and not a double-blind randomized clinical trial. However, since the McGill short-form questionnaire was completed at the same time as the HDRS-D-17 and Beck’s II scores for depression, the investigation was blind and the participants had already been randomized to psychobiotics or placebo; thus, the participants were interviewed blindly as to the treatment, and the non-blind analysis refers only to the division of the participants according to their depression status and not to their treatment. Third, the “non-common sense” finding that non-depressed participants demonstrated greater pain reduction following psychobiotic intervention would be accepted under the knowledge of the difference in the severity of disruption of the gut microbiota diversity in non-depressed and depressed participants. An analysis of the participants’ microbiota could potentially shed more light on this issue, but the purpose of our research protocol was purely clinical and completely non-invasive. Fourth, we have to mention that the study participants received two distinct chemotherapy regimens, the FOLFOX and the FOLFIRI, which, theoretically, have the potential to affect the treatment outcome; since both depression and pain are closely related to the gut microbiota composition, a possible disturbance of microbial diversity could alter the severity of depression or pain perception. Finally, the number of depressed individuals who were allocated to psychobiotic treatment presented a decline after a month of treatment because of the improvement in depression status; it would be of interest to be able to study them as a sub-group of former depressed individuals who became non-depressed after psychobiotics, especially in relation to pain intensity. However, the number of such cases is too small for statistical analysis.

## 5. Conclusions

Based on the above, we conclude that the psychobiotic formula used, taken as adjuvant to routine analgesics for a month, seems to ameliorate pain intensity during the treatment period—and possibly up to 2 months thereafter. This is evidence at a highly significant level in non-depressed individuals and to a lesser degree in depressed individuals. The value of this finding becomes greater when taking into account the sub-cohorts of non-depressed and depressed participants taking placebo: they not only do not enjoy an alleviation of pain intensity, qualitatively and quantitively, but also experience deterioration due to both the disease progression and the continuing detrimental effects of chemotherapy on both the body and the mood.

## Figures and Tables

**Figure 1 nutrients-17-02751-f001:**
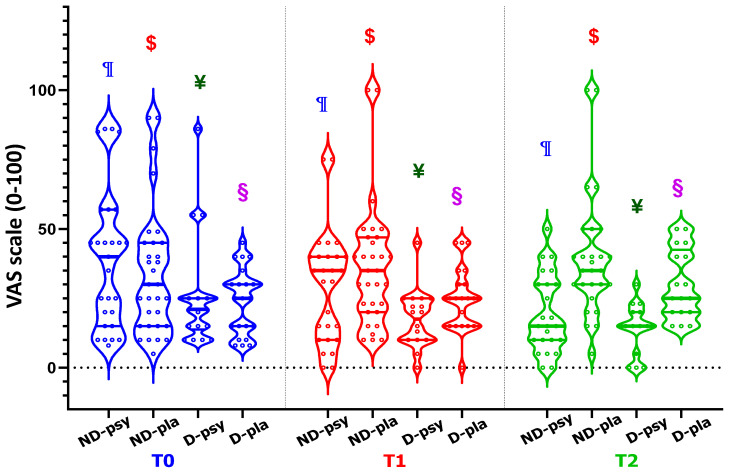
Pain intensity based on VAS score per study period and per group. ¶: *p* = 0.007, decrease; $: *p* = 0.045, increase; ¥: *p* < 0.001, decrease; §: *p* = 0.003, increase, between T0, T1, and T2 in the four groups, respectively—Friedman test. ND: non-depressed; D: depressed; psy: psychobiotics; pla: placebo. Horizontal lines represent median and IQR.

**Figure 2 nutrients-17-02751-f002:**
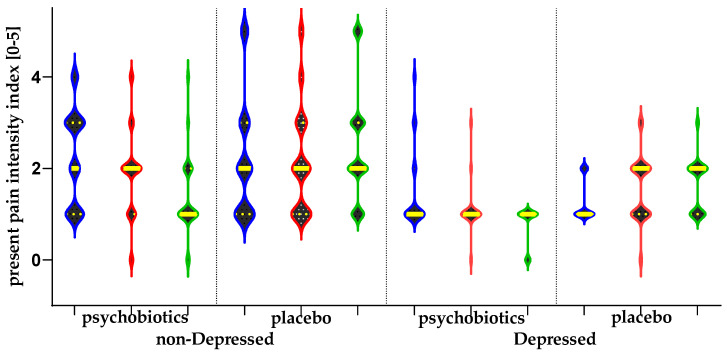
Illustration of the changes in the present pain intensity per study period. The horizontal yellow lines represent median and interquartile range (IQR). The blue violins represent T0 (baseline) assessment; the red T1 and the green the follow-up [T2].

**Figure 3 nutrients-17-02751-f003:**
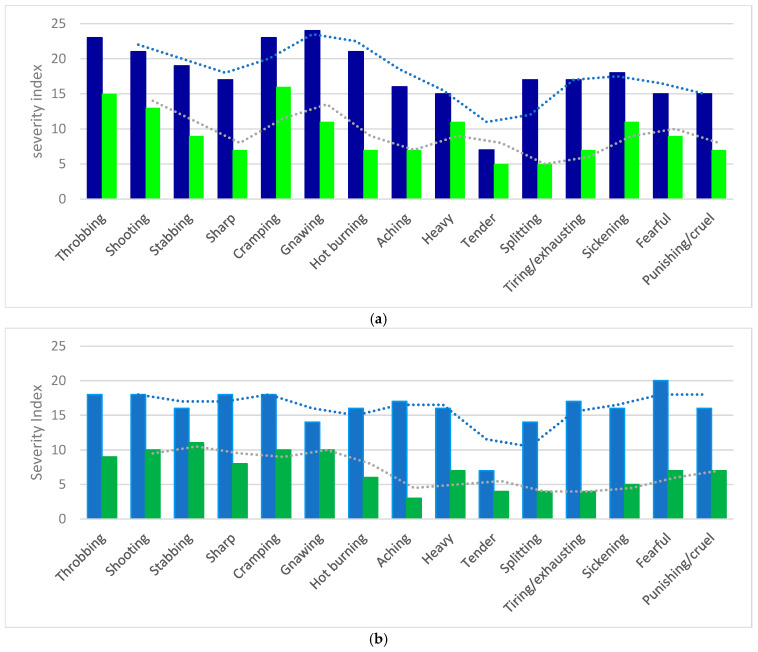
(**a**) Non-depressed and depressed participants taking psychobiotics; (**b**) non-depressed and depressed participants taking placebo. The first column of each pain descriptor, being taller (dark or light blue), represents the severity of pain (as a mean of the rating 0 to 3), having the described character in non-depressed individuals; the second column (bright or dark green) is for depressed individuals. All values are expressed at baseline—T0 (no treatment yet).

**Figure 4 nutrients-17-02751-f004:**
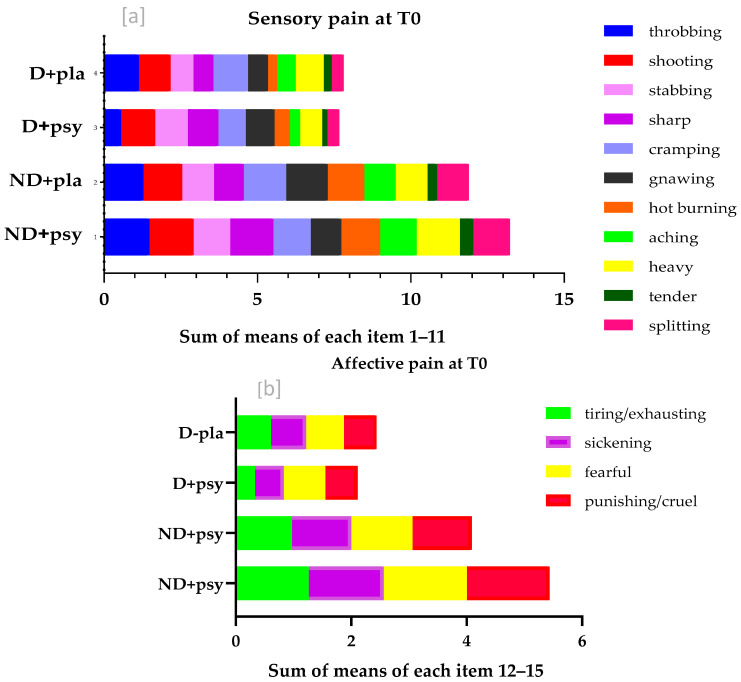
(**a**) illustrates the sum of means of the 11 items/descriptors referring to sensitive pain at T0, baseline (max value 33), and the differences of means of each of the 11 items per group of participants. (**b**) illustrates the sum of means of the 4 items/descriptors referring to affective pain at T0, baseline (max value 33), as well as the differences of means of each of the 4 items, per group of participants. The different length on the x-axis expresses the difference in the mean value of each descriptor, presented with a different color as well, to make the variances between groups most prominent. ND: non-depressed; D: depressed; psy: psychobiotics; pla: placebo.

**Figure 5 nutrients-17-02751-f005:**
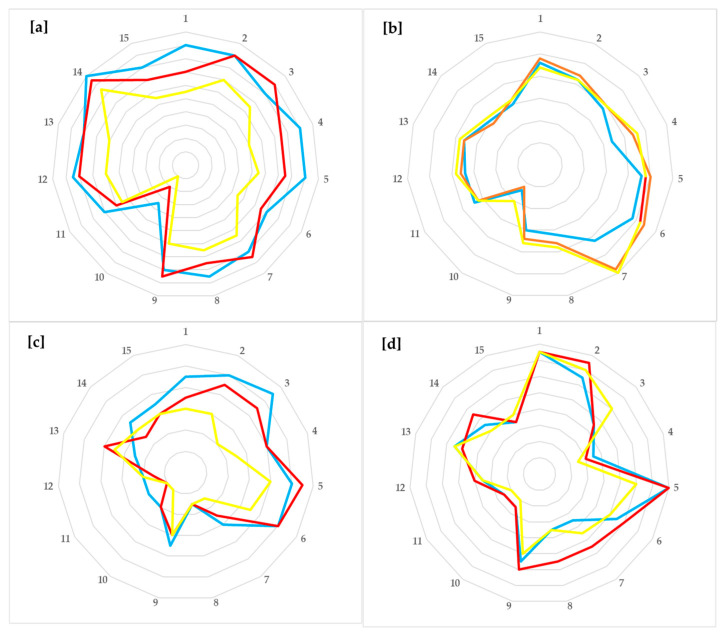
(**a**–**d**) Spider graph of the value fluctuations of each the 15 descriptors consisting the pain rating index per study period. The blue line represents T0—baseline values; the red line represents T1—after psychobiotics or placebo treatment termination; and the yellow line represents values at follow-up. (**a**,**b**) represent non-depressed individuals receiving psychobiotics and placebo, respectively; (**c**,**d**) represent depressed individuals receiving psychobiotics and placebo, respectively.

**Table 1 nutrients-17-02751-t001:** Pain-suffering participants with respect to psychobiotics or placebo treatment they had been previously allocated.

Treatment	Non-Depressed [n = 58]	Depressed [n = 41]	*p* *
** *Psychobiotics* **	27 [46.55%]	18 [43.90%]	0.794
** *Placebo* **	31 [53.45%]	23 [56.10%]	

Numbers (percentage of the corresponding group) express the numbers of participants per treatment × depression status; * chi-square test.

**Table 2 nutrients-17-02751-t002:** Pain intensity based on VAS score per group per study period.

Groups	Treatment	T0	T1	T2	*p* *
**Non-depressed** **(n = 58)**	** *Psychobiotics* **	38.9 [25.3]	29.3 [19.9]	19.3 [13.9]	**0.007**
	** *Placebo* **	34.1 [23.0]	36.1 [21.9]	39.7 [22.2]	**0.045**
***p* ****		**0.491**	**0.240**	**<0.001**	
**Depressed** **(n = 41)**	** *Psychobiotics* **	26.0 [19.9]	17.3 [10.3]	17.3 [10.3]	**<0.001**
	** *Placebo* **	24.3 [11.7]	24.6 [11.3]	30.0 [12.5]	**0.031**
***p* ****		**0.568**	**0.018**	**0.001**	

Numbers express the mean (SD) of the pain intensity score; * Friedman Test; ** Student *t*-test.

**Table 3 nutrients-17-02751-t003:** Present pain intensity per group and per study period.

Groups	Treatment	T0	T1	T2	*p* *	% Change T0–T1	% Change T1–T2	% Change T0–T2
**Non-Depressed** **(n = 58)**	** *Psychobiotics* **	2.3 [1.1]	1.9 [1.0]	1.4 [0.9]	**0.001**	3.0 [60.7]	−23.1 [38.8]	−17.4 [39.3]
	** *Placebo* **	2.2 [1.3]	2.2 [1.2]	2.5 [1.3]	**0.006**	0.4 [21.1]	22.2 [40.0]	24.7 [40.9]
***p*** ******		**0.526**	**0.582**	**0.001**		**0.022**	**<0.001**	**<0.001**
**Depressed** **(n = 41)**	** *Psychobiotics* **	1.5 [0.9]	1.1 [0.6]	0.8 [0.4]	**0.002**	−14.9 [26.3]	−22.6 [39.0]	−36.3 [40.2]
	** *Placebo* **	1.3 [0.5]	1.6 [0.7]	1.6 [0.6]	**0.010**	17.6 [39.3]	23.5 [47.2]	38.2 [48.5]
***p*** ******		**0.672**	**0.018**	**0.001**		**0.002**	**<0.001**	**0.003**

Numbers express the mean (SD) of the present pain intensity; * Friedman test; ** Student *t*-test.

**Table 4 nutrients-17-02751-t004:** Pain rating index per group and per study period (for all 15 items).

Groups	Treatment	T0	T1	T2	*p* *
**Non-depressed** **(n = 58)**	** *Psychobiotics* **	18.6 [13.2]	15.1 [11.1] ^#^	12.1 [8.7]	**<0.001**
	** *Placebo* **	16.0 [10.3]	16.5 [9.8]	19.3 [11.0]	**<0.001**
***p* ****		**0.558**	**0.719**	**0.022**	
**Depressed** **(n = 41)**	** *Psychobiotics* **	9.7 [7.9]	8.1 [6.1]	7.1 [6.2]	**0.106**
	** *Placebo* **	10.3 [5.8]	10.9 [6.3]	14.1 [7.8]	**0.002**
***p* ****		**0.420**	**0.133**	**0.008**	

Numbers express the sum of the severity index (rating from 0 to 3) of each of the 15 items of the McGill short-form questionnaire as mean (SD); * Friedman test; # *p* = 0.005 between T0 and T1; ** Student *t*-test.

**Table 5 nutrients-17-02751-t005:** Descriptive statistics of the sensory pain rating index [11 items] and the affective pain rating index [4 items] in non-depressed and depressed individuals receiving either psychobiotics or placebo. * Comparison between Psychobiotics and Placebo, ** Comparison within the same Group between the three time-points.

**Non-depressed**																					
			**T = T0**						**T = T1**						**T = T2**						*p*-Value **
Subtotal Score		Range	min	max	mean	SD	median	IQR	min	max	mean	SD	median	IQR	min	max	mean	SD	median	IQR	
Items 1–11	Psychobiotics	0–33	0	32	13.22	10.10	12.00	16.00	0	28	10.37	7.95	12.00	15.00	0	18	8.13	6.35	8.00	13.75	**0.004**
	Placebo		0	27	11.90	7.07	12.00	8.00	0	30	12.58	6.77	11.00	8.00	0	29	14.59	7.58	14.00	8.00	**<0.001**
*p*-Value *					**0.667**						**0.516**						**0.007**				
Items 12–15	Psychobiotics	0–12	0	11	5.44	3.77	5.00	7.00	0	11	4.70	3.54	5.00	7.00	0	10	3.96	3.03	3.00	4.00	**<0.001**
	Placebo		0	11	4.10	3.71	4.00	7.00	0	12	3.87	3.75	4.00	7.00	0	11	4.74	4.08	4.00	9.00	**0.123**
*p*-Value *					**0.165**						**0.372**						**0.535**				
**Depressed**																					
			**T = T0**						**T = T1**						**T = T2**						*p*-Value **
Subtotal Score		Range	min	max	mean	SD	median	IQR	min	max	mean	SD	median	IQR	min	max	mean	SD	median	IQR	
Items 1–11	Psychobiotics	0–33	0	21	7.67	5.53	6.50	5.50	0	16	6.33	4.73	6.00	7.00	0	16	5.33	4.64	5.00	10.00	**0.128**
	Placebo		0	15	7.83	4.47	9.00	7.00	0	15	8.22	4.81	10.00	7.00	0	19	10.53	5.79	10.00	8.50	**0.003**
*p*-Value *					**0.750**						**0.245**						**0.011**				
Items 12–15	Psychobiotics	0–12	0	10	2.11	3.07	0.50	4.00	0	7	1.78	2.44	0.50	3.25	0	5	1.73	2.09	0.00	3.00	**0.280**
	Placebo		0	9	2.43	2.68	2.00	4.00	0	10	2.70	2.77	2.00	4.00	0	10	3.59	2.92	4.00	2.50	**0.025**
*p*-Value *					**0.394**						**0.171**						**0.074**				

## Data Availability

The data that support the findings of this study are available on request from the corresponding author, K.K.

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
