# Peer review of "Beyond Analgesia: Psychobiotics as an Adjunctive Approach to Pain Management in Gastrointestinal Oncology—A *Post Hoc* Analysis from the *ProDeCa* Study"

_nutrients, 2025, doi:10.3390/nu17172751_

Round 1
Reviewer 1 Report
Comments and Suggestions for Authors
Dear authors, I have read the manuscript, and I send you my comments. It is not a clinical study? It is a post hoc analysis of a previous study; therefore, I think that it is a statistical study of a previously published study. Therefore, in my opinion, it is not a novelty and does not have a priority for publication.
1) Introduction: Line 163-168, it could be experimental protocol, not the aim of the study.
2) The experimental protocol is very difficult to understand.
Author Response
REVIEWER 1
Dear authors, I have read the manuscript, and I send you my comments. It is not a clinical study? It is a post hoc analysis of a previous study; therefore, I think that it is a statistical study of a previously published study. Therefore, in my opinion, it is not a novelty and does not have a priority for publication.
Response: Thank you for your comments!
This article is part of a multifaceted protocol concerning the action and benefits of psychobiotics treatment regarding depression, anxiety, stress, pain, emotion regulation and quality of life in patients undergoing chemotherapy after surgery for gastrointestinal cancers. The first publication [Tzikos, G et al, Nutrients, 2025; 17(5), 857] concerned the depression/anxiety experienced by 99 of the 266 participants and analyzed the benefits of psychobiotics in significantly improving their mood status.
Our manuscript, submitted and under review, deals with the assessment of pain and the significantly beneficial role of psychobiotics in the amelioration of pain intensity; it constitutes the second part of our research protocol, which could not be presented in any other way than as a post-hoc analysis, since the patients’ randomization had already been determined for the previous publication.
While the statistical methodology used in this study may be standard, the contribution of the study is not statistical in nature—it is a clinical research study [unfortunately not RCT for the above explained reason] based on the interplay between the gut-brain axis, psychobiotics, depression, and pain perception.
We thus strongly argue that this post-hoc analysis is of great importance for four reasons:
* First, there is no other similar study on psychobiotics treatment in a population undergoing a course of chemotherapy after surgery for gastrointestinal cancer;
* Second, because these patients constitute two groups - those with depression and those without - the effect of psychobiotics on pain can be studied separately in those with or without depression, enabling the correlation of the improvement in pain intensity in relation to the depression status;
* Third, because of the finding that the non- depressed patients were shown to be more beneficially affected from the psychobiotics treatment than the depressed; and
* Finally, and most importantly, because of the unexpected finding that depressed participants experienced less pain compared to the non-depressed; this being in contrast to general thinking that depressed individuals experience more pain, apparently because they exaggerate their reactions. We try to interpret this finding in “Discussion” with the help of the “gate-control” theory and other references [please see the manuscript, lines 585-602.
Based on the above, we consider that your suggestion that our study be rejected because it has nothing new to present and is nothing more than a statistical analysis and therefore of no interest, should perhaps be revised in the light of the above.
1) Introduction: Line 163-168, it could be experimental protocol, not the aim of the study.
Response: In no way can this work be considered an experimental protocol. It is a post-hoc analysis, significantly original in the field of psychobiotics.
2) The experimental protocol is very difficult to understand.
Response: The “research protocol” and not the “experimental protocol” of this study, is presented briefly, as is usual in post-hoc analyses in order to avoid plagiarism, and a full reference to the original study is given [Tzikos, G et al, Nutrients, 2025; 17(5), 857].
For your convenience, we presented again in other words: two groups of depressed patients had already been allocated to receive psychobiotics or placebo and two other groups of non-depressed patients also received psychobiotics or placebo. The allocation to psychobiotics or placebo groups had been performed prior to participants assessment according to HDRS-D-17 score as depressed and non-depressed. In each group, the number of those who were in pain was recorded and their course monitored and assessed according to the "treatment" [psychobiotics or placebo]. And finally, a comparison was made both within each group and between the groups.

Reviewer 2 Report
Comments and Suggestions for Authors
The manuscript entitled “Beyond Analgesia: Psychobiotics as an Adjunctive Approach to Pain Management in Gastrointestinal Oncology – A Post-Hoc Analysis from the ProdeCa Study” is an interesting paper on the potential analgesic effects of probiotics in pain control in cancer patients receiving chemotherapy. Nevertheless, some remarks mus be carried out and discussed:
- The post-hoc nature of the presente analysis raises important methodological considerations. The authors are encouraged to discuss the potential impact of this design on the validity and conclusions.
- The observation that non-depressed patients reported higher pain scores than those with depression is against common sense. This result should be more thoroughly interpreted, ideally with support from existing literature.
- Increasing the sample size in future studies would be advisable to strengthen the statistical power.
- Considering that the gut–brain axis is often more profoundly altered in individuals with depression, the finding that non-depressed patients demonstrated greater pain reduction following psychobiotic intervention appears paradoxical. Does this suggest different mechanisms of action or patient-specific factors influencing treatment response?
- Given the distinct profiles of the FOLFOX and FOLFIRI chemotherapy regimens, the potential influence of treatment heterogeneity on the outcomes should be addressed. Could this difference alter the data and conclusions?
Author Response
REVIEWER 2
The manuscript entitled “Beyond Analgesia: Psychobiotics as an Adjunctive Approach to Pain Management in Gastrointestinal Oncology – A Post-Hoc Analysis from the ProdeCa Study” is an interesting paper on the potential analgesic effects of probiotics in pain control in cancer patients receiving chemotherapy. Nevertheless, some remarks must be carried out and discussed:
- The post-hoc nature of the present analysis raises important methodological considerations. The authors are encouraged to discuss the potential impact of this design on the validity and conclusions.
Response: Thank you for your comment, giving us the opportunity to highlight the advantages as well as the new findings arising from our study.
Our manuscript, submitted and under review, deals with the assessment of pain and the significantly beneficial role of psychobiotics in the amelioration of pain intensity. It constitutes the second part of our research protocol, so it could not be presented in any other way than as a post-hoc analysis, since the patient randomization had already been determined for the previous publication.
However, it should be noted that the randomization of participants to treatment [psychobiotics or placebo] had been done initially, and then the blind evaluation with the psychometric tests for depression/anxiety, emotional status, followed by the 15-item questionnaire of McGill for pain. Thus, there can be no claim for bias in participant assessment.
- The observation that non-depressed patients reported higher pain scores than those with depression is against common sense. This result should be more thoroughly interpreted, ideally with support from existing literature.
Response: Thank you for your comment. It is true that common sense dictates that depressed individuals will report higher pain severity. It is also recognized by some authors [Wiech, K and Tracey, I. 2009 and Schreier, A.M.; et al 2019 [ref 17, 18] that “anxiety and depression, being common comorbidities in cancer patients can significantly amplify pain perception” [lines 96-98].
However, we found just the opposite - that the prevalence of reported pain in depressed cohorts is lower. This is in agreement with the findings of some other authors. We have already referred to the “gate-control” theory - please read the paragraph - lines 585-602. To further strengthen the interpretation of our results, we have added the following paragraph [please see lines 603-622]:
“Based on or affected by the “gate-control theory”, other authors further analyze and explain the findings either as due to perception and reporting bias or simply as due to differences in coping mechanisms. Depressed individuals may underreport their physical symptoms, including pain, due to general apathy, emotional blunting, or reduced interoceptive awareness that often accompanies depressive states. Others suggest that depressed individuals may develop pain tolerance or resignation as a coping mechanism, as opposed to non-depressed individuals who might be more reactive or attentive to pain sensations [Geisser ME, et al. Negative affect, self-report of depressive symptoms, and clinical depression: Relation to the experience of chronic pain. Clin J Pain. 2000;16(2):110]. This option is also supported by the biochemical theory of depression, which suggests that a neurochemical imbalance of monoamines, as is serotonin, underlies depression and, since these neurotransmitters are likely to play a key role in modulatory pain pathways, this may lead to altered pain perception [Thompson T, et al. Is Pain Perception Altered in People with Depression? A Systematic Review and Meta-Analysis of Experimental Pain Research. J Pain. 2016 Dec;17(12):1257-1272]. Additionally, a preliminary meta-analysis of 6 experimental studies [Dickens C, et al: Impact of depression on experimental pain perception: A systematic review of the literature with meta-analysis. Psychosom Med 65:369-375, 2003] reported decreased pain in depressed groups relative to healthy control groups. This finding has become more significant after a new meta-analysis – the largest and most comprehensive to date – of 25 studies [567 depressed and 587 healthy participants] investigating responses to experimentally-induced pain, which revealed a higher overall pain threshold, meaning reduced pain in the depressed in comparison to healthy participants [Thompson T, et al. Is Pain Perception Altered in People with Depression? A Systematic Review and Meta-Analysis of Experimental Pain Research. J Pain. 2016 Dec;17(12):1257-1272]”.
- Increasing the sample size in future studies would be advisable to strengthen the statistical power.
Response: Thank you for your suggestion. We begin our protocol with 264 patients [sample size calculation: 132 patients in each arm], aiming to reduce the prevalence of depression by 50%. In the present post-hoc analysis we have the two groups [psychobiotics or placebo treated] further sub-divided into depressed and non-depressed, which unfortunately sub-divides the number of participants per group. However, our comparisons are of significance difference.
- Considering that the gut–brain axis is often more profoundly altered in individuals with depression, the finding that non-depressed patients demonstrated greater pain reduction following psychobiotic intervention appears paradoxical. Does this suggest different mechanisms of action or patient-specific factors influencing treatment response?
Response: It has been well recognized for years that there is a close relationship between gut microbiota and pain perception, as well as that psychiatric disorders are clearly related to alterations in the gut microbiota composition and, more specifically, in the low-diversity microbiome or pathobiome, [Menni AE, et al. Rewiring Mood: Precision Psychobiotics as Adjunct or Stand-Alone Therapy in Depression Using Insights from 19 Randomized Controlled Trials in Adults. Nutrients. 2025 Jun 17;17(12):2022] [Fyntanidou B, et al. Probiotics in Postoperative Pain Management. J Pers Med. 2023 Nov 25;13(12):1645], leading to the [unproved question] “chicken and egg” dilemma. On the other hand, the non-depressed individuals could be considered to have a less disturbed microbiota [only disease-related] and thus present a greater improvement in pain perception after psychobiotics treatment, contrary to the depressed, also suffering pain, who could be considered as having a highly disturbed microbiota, which may require greater effort to restore by measures such as higher doses of pro- and/or psychobiotics or even a longer treatment period. In the light of their depression severity, they are likely to present a lower degree of improvement.
We understand that there is still a huge gap in our knowledge concerning the role of psychobiotics in this field, since our knowledge is still relatively poor merely regarding the interaction between depression and pain. An analysis of participants’ microbiota could potentially shed more light on this issue, but the purpose of our research protocol was purely clinical, and completely non-invasive.
Thus, we have simply added the following sentence to the study limitations [lines 636-643]:
“Third, the “non- common sense” finding that non-depressed participants demonstrated greater pain reduction following psychobiotic intervention would be accepted under the knowledge of the difference in the severity of disruption of the gut microbiota diversity in non-depressed and depressed participants. An analysis of the participants’ microbiota could potentially shed more light on this issue, but the purpose of our research protocol was purely clinical, and completely non-invasive”.
- Given the distinct profiles of the FOLFOX and FOLFIRI chemotherapy regimens, the potential influence of treatment heterogeneity on the outcomes should be addressed. Could this difference alter the data and conclusions?
Response: We acknowledge that our patients received two different chemotherapy regimens, which may have resulted in bias in the final results, although the aim of this post-hoc analysis was pain improvement by means of psychobiotics and not the effectiveness of chemotherapy.
Moreover, from the limited relevant literature, it would seem that the FOLFOX and FOLFIRI chemotherapy regimens do not appear to affect the gut microbiota [Unrug-Bielawska K, et al. Comparative Analysis of Gut Microbiota Responses to New SN-38 Derivatives, Irinotecan, and FOLFOX in Mice Bearing Colorectal Cancer Patient-Derived Xenografts. Cancers (Basel). 2025 Jul 7;17(13):2263], but rather the effectiveness of the chemotherapy depends on the existing gut microbiota [Li J, et al. Composition of fecal microbiota in low-set rectal cancer patients treated with FOLFOX. Ther Adv Chronic Dis. 2020 Feb 26;11:2040622320904293] [Fang Q, et al. Gut microbiota derived DCA enhances FOLFOX efficacy via Ugt1a6b mediated enterohepatic circulation in colon cancer. Pharmacol Res. 2025 Mar;213:107636] [Zhu X, et al. A tumor microenvironment-specific gene expression signature predicts chemotherapy resistance in colorectal cancer patients. NPJ Precis Oncol. 2021 Feb 12;5(1):7]. Thus, in recent years, many studies have been oriented towards modulation of the gut microbiota to improve chemotherapy drug efficacy, and many gut microbes have been recognized as positively or negatively affected it. [Li J, et al. Composition of fecal microbiota in low-set rectal cancer patients treated with FOLFOX. Ther Adv Chronic Dis. 2020 Feb 26;11:2040622320904293.]
Unfortunately, it was impossible to split the participants in our study into sub-groups based on their chemotherapy regimen, since such a division would lead to small-number groups and possibly to other bias. In the same manner, it was impossible to analyze the participants’ gut microbiota since the purpose of our research protocol was purely clinical, and completely non-invasive.
Thus, we have added the following sentence in the study limitations – please see in the last paragraph of the Discussion section [lines 643-648]:
“Fourth, we have to mention that the study participants receive two distinct chemotherapy regimens, the FOLFOX and the FOLFIRI, which, theoretically, have the potential to affect the treatment outcome; since both depression and pain are closely related to the gut microbiota composition, a possible disturbance of microbial diversity could alter the severity of depression or pain perception”.

Reviewer 3 Report
Comments and Suggestions for Authors
This manuscript studied the psychobiotic effect on pain management in cancer patients. The entire manuscript is based on the pain index completed by patients, which is inherently subjective, and lacks a mechanism study or explanation. Therefore, I believe this manuscript requires further work.
There is also some confusion about the results:
3.3 Table 2: How are these numbers calculated? By mean? By percentage? What are the real numbers that have changed from T0 to T2?? The results are very confusing.
Line 321-323: In the introduction, the pain is linked with depression. So why did the results show that non-depressed individuals experience higher intensity of pain? Any explanation?
Table 3: Are these numbers the pain index? I think the figure clearly shows the pain index. If the authors could also include the percentage of participant change in the Table, it would be more helpful.
Figure 3: What is the Y-axis?
Overall, this manuscript needs more critical thinking and organization, even though it is worth studying.
Author Response
REVIEWER 3
This manuscript studied the psychobiotic effect on pain management in cancer patients. The entire manuscript is based on the pain index completed by patients, which is inherently subjective, and lacks a mechanism study or explanation. Therefore, I believe this manuscript requires further work.
Response: Thank you for your comment! We accept that our paper is based on a subjective questionnaire, the short form, 15-item questionnaire of McGill. However, this test is used worldwide for clinical studies, validated for many countries and remains, to date, the most widely used test for pain assessment. On the other hand, we, as physicians, have no other choice but to rely on patient self-reports, which, in our study have the inherent advantage of being consecutive measurements, thus reducing the possibility of bias.
We refer to this disadvantage in the last paragraph of the Discussion, in the study limitations – Lines 623-636.
There is also some confusion about the results: 3.3 Table 2: How are these numbers calculated? By mean? By percentage? What are the real numbers that have changed from T0 to T2?? The results are very confusing.
Response: Table 2 presents data [mean [SD]] on Pain intensity, based on the Visual Analogue Scale [VAS – max 100 points], per group per study period; the higher the score the greater the pain intensity, as perceived by the patient. Just below Table 2 there is a footnote explaining that "Numbers express the mean [SD] of the pain intensity score”; and that statistical analyses within the groups [*] have been performed with the Friedman Test; while that between groups [**], with the Student’s t-test, as also referred in the Statistics section [2.5. – Lines 257-274]. This Table demonstrates how the mean [SD] VAS score fluctuates over time within each group, allowing a clearer understanding of the trend in pain perception over the study course.
The real numbers you are asking for – that is the pain intensity score of each participant belonging to each of the four groups and for each study period – are presented in Figure 1 [Lines 329], just below Table 2. In the violin-form columns each open circle represents the VAS score of each individual, with the horizontal lines expressing median [IQR].
Lines 321-323: In the introduction, the pain is linked with depression. So why did the results show that non-depressed individuals experience higher intensity of pain? Any explanation?
Response: Thank you for this thoughtful observation. In lines 89-102 we present the option of Wiech, K and Tracey, I. 2009 and Schreier, A.M.; et al 2019 [ref 17, 18] that depression status augments the pain experience. We thus wrote: "Cancer pain, affecting nearly 66% of cancer patients is not merely a somatic experience but a complex phenomenon involving emotional and psychosocial dimensions...From this point of view, the close relationship of emotional response to any sensory dimension, and vice versa, suggests that anxiety and depression, being common comorbidities in cancer patients can significantly amplify pain perception through top-down modulation in the central nervous system”.
However, we found the opposite, that the depressed individuals experienced less pain. Some authors support this idea: depressed patients may underreport physical symptoms, including pain, due to general apathy, emotional blunting, or reduced interoceptive awareness that often accompanies depressive states or may be due to development of pain tolerance or resignation as a coping mechanism [please see Lines 603-622 – we have added a paragraph, as an answer to a comment of another reviewer]. Besides this, we support and explain this option in the Discussion, using the “gate-control” theory - please read the previous paragraph - lines 585-602.
Table 3: Are these numbers the pain index? I think the figure clearly shows the pain index. If the authors could also include the percentage of participant change in the Table, it would be more helpful.
Response: This is a useful comment. The numbers express the mean [SD] of the present pain intensity index, being a categorical scale ranging from 0 to 5. At your suggestion, we have added to the Table the percentage of participant change, in order to provide a more comprehensive view of the clinical evolution of pain perception throughout the study. Please find the new Table 3 within the text, Line 342.
Figure 3: What is the Y-axis?
Response: Thank you again for your useful comment. The Y-axis in both Figures [3a and 3b] represent the sum of the severity index assessed by the McGill short form questionnaire [rating from 0 to 3 and expressed as mean [SD]] of each individual item-descriptor, which are also presented in Supplementary Table 2. Please find the new figures with a legend for the Y-axis.
Overall, this manuscript needs more critical thinking and organization, even though it is worth studying.
Response: We feal that if read with care and attention, and taking note of our responses about this article is both clear and deals with the subject critically, which provide an interesting and relevant contribution to this field, which we agree is worth studying.

Round 2
Reviewer 1 Report
Comments and Suggestions for Authors
Dear Authors,
I have read again your manuscript and I have the same comments respect my previous evaluation. This is not a clinical study or review or case report but a statistical study of previous published results
Reviewer 3 Report
Comments and Suggestions for Authors
With the revision, the manuscript hits the publisher standar